# From Learning Plot to Main Field: Scaling-Out Soil Health Innovations in Malawi

Frank Tchuwa [1,*], Kate Wellard [2], John Morton [2], Daimon Kambewa [1], Daniso Mkweu [1] and Wezi Mhango [1]

[1]  Bunda College, Lilongwe University of Agriculture and Natural Resources,
    Lilongwe P.O. Box 219, Malawi; dkambewa@luanar.ac.mw (D.K.); mkweudaniso@gmail.com (D.M.);
    wmhango@luanar.ac.mw (W.M.)
[2]  Natural Resources Institute, University of Greenwich, Chatham ME4 4TB, UK;
    k.wellard@greenwich.ac.uk (K.W.); j.f.morton@greenwich.ac.uk (J.M.)
*   Correspondence: ftchuwa@luanar.ac.mw; Tel.: +265-996-545-274

**Abstract:** Farmer-centred approaches are applied to engage smallholder farmers in agricultural research and development with the purpose of identifying and scaling out context specific innovations. Understanding the underlying processes that influence the decision of smallholder farmers to scale-out innovations is, therefore, paramount to effective farmer-led research and development programmes. This study analysed how smallholder farmers in rural Malawi were involved in evaluating soil health management options as well as how they scaled-out the lessons from the learning plots to their main farms. Data were collected through observations and face-to-face interviews in 109 learning plots and 197 main fields managed by farmers who participated in interventions that applied farmer-centred approaches. The findings reveal that farmers' capacity to engage in systematic experimentation depended on their knowledge of basic research principles and their social capital (bridging and bonding). Farmers observing and interacting in the learning plots formed different perceptions about the performance of the tested options. The variations in the perceptions were associated with biophysical (plot characteristics) and socioeconomic factors (time of planting). Likewise, variations were observed in the way farmers scaled-out the tested options. Whilst some farmers integrated many different options (>3), others applied few options in their main fields (<3). The majority of farmers adapted the options to suit their contexts. Farmers' decision to scale-out options was associated with their perceived benefits of the options, gender, and wealth status. The study findings have implications for research and development programmes that use farmer-centred approaches to push for adoption of blanket recommendations.

**Keywords:** soil health innovations; scaling-out; smallholder farmers; experiential learning; social learning; transformative learning

## 1. Introduction

In Sub-Saharan Africa, agricultural interventions aim at improving smallholder farm productivity by facilitating farmers' behaviour change towards more productive and sustainable farming practices. The goal is to help farmers attain optimum crop yield by learning and applying appropriate innovations. According to FAO [1] and the World Bank [2], the availability and application of new ways of production enable small-scale producers to overcome a wide and often complex range of farm problems, including climate change, pests and diseases as well as the loss of soil health. However, the challenge for implementers of agricultural interventions is how to provide effective learning opportunities to farmers that lead to desired behavioural changes. The literature provides theories that agrarian adult educators can apply when designing and implementing effective learning opportunities, including experiential, transformational and social learning theories. These theories explain how learning occurs among adults as well as how it can be enhanced.

Experiential learning theory posits that learning occurs through a process in which knowledge is created by transforming experiences [3,4]. For agricultural non-formal education programmes, the process in this experience-based learning starts with farmers engaging in a concrete experience. For instance, involvement of a producer group in testing a new method of planting a legume crop on a small learning plot (could be a demonstration or experimental plot). The farmers make observations and reflect on their experiences with the innovation, and then create abstract concepts about how to apply this new practice of planting in their contexts. Finally, they actively experiment with the learned planting pattern on their farms. As they try it out in their fields, a new concrete experience is created, thereby providing a basis for another learning cycle.

Transformational learning theory [5,6] argues against the learning opportunities in which the participants are restricted to listening and accepting facts given by the educators. Instead, it asserts that learners should be liberated to critically reflect on themselves and their social, political, cultural, and economic contexts. This reflection process helps individuals change their perspectives and have a deeper understanding of the problems that affect their lives. The outcome of such a consciousness-raising process is learners who are empowered to take actions that change their world.

The application of transformative agricultural adult-learning initiatives entails shifting from top-down approaches. In this case, addressing farm productivity constraints by pushing for farmers' reproduction of instructions that have been developed and disseminated through formal research and extension institutions is not the focus. What is critical then is giving farmers the freedom to reflect and solve their farming problems critically. With a transformation perspective, it is believed that the small-scale producers will apply appropriate farming principles and adapt innovations to suit their context.

Finally, social learning theory emphasises the acquisition of new knowledge and practices through the interactions between individuals, groups or communities [7,8]. At the individual level, learning occurs in human beings as they observe the behaviour of other individuals, often perceived as models in a particular social context [9]. This type of learning takes place among farmers when they interact and attentively observe, encode (remember) and imitate what they perceive as positive farming practices from others. In a typical agrarian setting, the models could be progressive farmers, fellow group members and various change agents operating in the communities. Social learning theorists argue that learning does not always lead to a change in behaviour. There are cases where learners acquire the competence to imitate or reproduce the observed behaviour but fail to sustain its exhibition due to the unavailability of adequate conditions of reinforcement or rewards. Likewise, the farmers may have learned several new farming practices but choose to continue applying only those that they perceive as suitable and beneficial to their households.

Despite the learning theorists contending that learning among adults is neither mechanical (linear) nor one-way, studies assessing the effectiveness of applying farmer-centred approaches in facilitating behaviour change among farmers tend to focus on adoption. Most of these adoption studies look for evidence of farmers reproducing technologies and blanket recommendations from research and development institutions. For example, some studies looked at how farmers participating in the Local Agricultural Research committees had adopted improved bean varieties [10] and pest management practices in cocoa production [11]. Another study on farmer researchers in Northern Malawi also gathered evidence on adoption of legume crops and incorporation of crop residues into the soil [12]. The literature also contains several adoption studies on improved technologies promoted to farmers participating in farmer-centred approaches [13–17].

The study presented in this paper aimed at addressing the question of how participation in interventions that apply farmer-centred approaches contributes to farmer learning and scaling-out of soil health management options. Specifically, the study focused on analysing the processes that occur from the moment farmers test different soil health management options in the learning plots to the point they decide to apply the tested options

in their main farms. Based on the definition provided by Doran and Zeiss [18], the term soil health management option is used in this paper when referring to innovation with the potential to balance the interaction of soil chemical (e.g., nutrients), physical (e.g., texture and structure) and biological (e.g., plants and animals) properties.

Duegd and associates [19] proposed the testing of a "praxeology" in which Participatory Rural Appraisal (PRA) and Participatory Technology Development (PTD), evaluation and feedback were employed to engage farmers in soil health innovation. Later, scholars noted that various participatory approaches were being used to promote technologies and practices related to soil health management in Sub-Saharan Africa [20]. Likewise, the focus of this paper is on different participatory approaches applied in interventions that supported soil health innovation in diverse contexts. The approaches include the Lead Farmer (LF), Farmer Field School (FFS), Farmer Research Team (FRT) and Farmer Research Network (FRN). These approaches were applied in agricultural interventions in Kandeu, Mkanakhoti and Zombwe Extension Planning Areas (EPA) in Malawi. Over 90% of smallholder farmers in these areas grow maize as a staple crop. However, productivity is low (less than 2000 kg/ha) due to biophysical factors such as unpredictable rainfall pattern, pest and disease pressure, cultivation of marginal and unsuitable land, and soil degradation [21,22]. The rapid population increase and the consequent small landholdings, coupled with the pro-maize policies, markets and institutions, have pushed farmers into practicing maize monoculture [23].

The Ministry of Agriculture defines a Lead Farmer as an individual farmer who is elected by the community to perform technology specific farmer-to-farmer extension and is trained in the technology. These expert farmers are expected to link other farmers in the community (known as follower farmers) to information on agricultural innovations, especially in contexts where the ratio of extension workers to farmers is low [24,25]. A Farmer Field School is a collection of farmers who get together to study a topic related to their farming problems and needs. Under this experiential learning approach, farmers are provided with a learning opportunity through which they acquire basic agricultural and management skills that make them experts in their own farms [26]. A Farmer Research Team is a platform where a selected team of farmers take charge of the agricultural research process that benefits both them and their community. The farmers drive the experimentation process through which technologies are evaluated and adapted. The findings from the experimentation are then reported to the community, where suitable recommendations are identified and disseminated to the broader farming community [27]. A Farmer Research Network is an association of farmer groups working together with research and development organizations to facilitate sharing of information and data as well as access to technical, institutional and financial support. The members of the network collaborate to conduct high-quality and credible research on mutually agreed topics to address collectively agreed constraints [28,29].

## 2. Methods

### 2.1. Data Collection

Data collection in the learning plots involved observations and face-to-face interviews. From March to May 2017, the research team visited 109 learning plots where farmers were testing different soil health management options in the study areas. A checklist guided the observations on the plots. The visits to the learning plots were restricted to only those farmers participating in interventions of interest to this study (i.e., exclusive to Lead farmer, FFS, FRT and FRNs working on soil health-related innovations). After observing the options in the plots, the individual farmers managing the learning plots in the FRTs and FRNs were then interviewed on-site using a semi-structured questionnaire. Group interviews were conducted where farmers collectively managed a single plot (i.e., lead farmers and FFS).

Out of the 109 learning plots visited 51 were managed by farmers in the FRN (34 females and 17 males), 43 in the Lead Farmer (24 females and 19 males), 11 in the FRT (7 females and 4 males) and 4 in the FFS. The number of plots from the FRT approach

was smaller than anticipated (35) because the majority of the members indicated they had stopped testing soil health options on small plots. They reported that they were now practicing some options in their main fields. Since each FFS was organised around a single school plot where a group of farmers interacts, the 4 plots visited represent all the FFS selected in this study.

Data collection in the main fields was conducted from April to May of 2017 and involved 197 farmers who were participating in the same interventions (Lead farmer, FFS, FRT and FRN) and were managing the learning plots (Table 1). Again, the exercise started with observing the fields using a checklist and then interviewing the farmers using a semi-structured questionnaire. The observations and interviews in the learning plot and main field were conducted after the purpose and use of data was explained and consent was given by the farmer (s).

**Table 1.** Distribution of the farmers in the main field study.

| Intervention | EPA | Sample Frame | Sample Size | | | % of Sample Frame |
| --- | --- | --- | --- | --- | --- | --- |
| | | | Female | Male | Total | |
| LF | Kandeu | 55 | 30 | 24 | 54 | 96 |
| FFS | Mkanakhoti | 56 | 55 | 1 | 56 | 100 |
| FRT | Zombwe | 35 | 19 | 16 | 35 | 100 |
| FRN | Kandeu, Mkanakhoti, Zombwe | 55 | 34 | 18 | 52 | 92 |
| Total | | 201 | 138 | 59 | 197 | 98 |

EPA = Extension Planning Area, LF = Lead Farmer, FFS = Farmer Field School, FRT = Farmer Research Team, FRN = Farmer Research Network.

*2.2. Type of Data*

During the visits, farmers were asked to indicate how they implemented and interacted in the learning plots. The focus was on the types of events, the frequency of interaction and the actors who attended such events. Various plot characteristics were then recorded together with the farmer(s). They included plot size, number of plots, soil type, slope, moisture condition, weeds, pests and diseases, as well as the types of soil health options tested. Finally, farmers were asked to evaluate the performance of soil health options tested in the plots. Farmers' opinions of the options were then recorded based on a 5-point scale (1 = very poor, 2 = poor, 3 = fair, 4 = good, 5 = very good).

In the main fields, farmers were first asked to indicate the soil health options that they had learned as a result of participating in the implementation of the learning plots. Then, the observations and recording of the types and number of options applied in their fields followed. The observations placed emphasis on identifying the integration and changes made to different options in the main field. In addition, collected were data on socioeconomic characteristics of the participants. These characteristics included farmers' perceptions of the benefits and constraints to applying the options in their fields, as well as their competence in applying the new soil health farming practices. Other socioeconomic variables collected included sex and wealth status of the farmers.

Based on a participatory wealth ranking exercise conducted in the study areas (in 2016), farmers were categorised as the "better-off" who could afford to purchase mineral fertilizer, owned a modern house (burnt brick wall and iron sheet roofed) and a wide range of assets (communication-radio and cell phone, livestock-goats and pigs and farm tools-watering can). They had at least 5 years of primary education. The "poor" who could not purchase fertilizer mainly owned a burnt brick wall house but with grass-thatched roof, chickens and a hand hoe. This category of farmers had attended not more than 5 years of primary education. Finally, the "very poor" who also could not afford a bag of fertilizer owned a mud wall house with a grass thatched roof. Even though they owned a hand hoe, very few of them owned chickens. Most of the farmers in this category had not attended primary education.

### 2.3. Data Analysis

Content analysis was used to analyse the qualitative data. This data processing technique involved identifying the themes and then establishing the relationships between patterns by reading through the farmers' responses to open-ended questions. The analysis of the qualitative data was conducted using the ATLAS.ti software (version 7.5.7). Processing of the quantitative data involved generating the descriptive statistics (frequencies and means), using the STATA software (version 13). Regression analysis was run to establish the association between the dependent variable and independent parameters.

In the social sciences, the standard analytical techniques that were used to determine the relationships between the dependent and a group of independent variables were linear, logistic, ordinal and multinomial regressions. The choice of the regression depends on the measurement scale of the outcome variable. Linear regression is suitable for continuous data. The categorical and dichotomous outcome variable was analysed using logistic regression. When the categories were ranked, the ordinal regression applied. When it was nominal with more than 2 groups, the multinomial regression was chosen [30]. In this study, multiple logistic regression was run to establish the sources of variations in the scores given by farmers regarding the response of maize yield to soil health options [31,32]. This binary regression suited the data since the dependent variable (Y) was dichotomous (1 = good maize response, 0 = poor maize response). The dependent variable (Y) was the natural log of the probability of good maize response to the soil health options (P) divided by the probability of a poor maize performance $(1 - P)$ by using any of the predictor variables (X). The independent variables used in the regression included the biophysical and management factors. Below is the specification of the logistic regression used in the study.

$$\ln [p/(1 - p)] = \alpha + \beta X + e$$

The above equation is simplified as shown below:

$$Y_1 = \ln (P/1 - P) = f (X_1, X_2, X_3, X_4, X_5) + e$$

where Y is the outcome variable, X is the explanatory factor, p is the probability, $\alpha$ is the intercept and $\beta$ is the regression coefficient for X. The symbol e is the error term.

The coefficients generated from the logistic equation shown above represent the log odds of an event occurring over those of the event not happening. However, the log odds do not provide a practical interpretation of the results. Therefore, in this study, the coefficients from the logistic regression were converted to odds ratio by the exponential function. The Stata command (.logistic) was run to generate the logistic regression parameters. Table 2 below shows the dependent and independent variables included in the logistic regression.

**Table 2.** Definition of variables in the multiple logistic regression.

|  | Variables | Units of Measure |
|---|---|---|
| Y | Maize response to soil health options | 0 = Poor, 1 = Good |
| $X_1$ | If the plot had loamy soils | 0 = No, 1 = Yes |
| $X_2$ | If the plot was affected by run-off | 0 = No, 1 = Yes |
| $X_3$ | If the plot was in waterlogged conditions | 0 = No, 1 = Yes |
| $X_4$ | If the plot was affected by moisture stress | 0 = No, 1 = Yes |
| $X_5$ | Time of planting | 0 = Late, 1 = On time |

The ordered logistic regression was run to analyse the factors associated with the variations in the number of soil health options applied by the farmers [33]. This regression was applied to this study since the dependent variable (Y) had ordinal data. The categories ranked as 1 = low integration of soil health options (when a farmer applied 0 to 2 options), 2 = moderate integration (farmer applied 3 to 4 options) and 3 = high integration (farmer applied more than 4 options). The predictor variables in the model were the socioeconomic

characteristics of farmers who participated in interventions that applied farmer-centred approaches. The specification of the ordered logistic regression run during the analysis is shown below.

$$\text{Log } \{Y(x)/(1 - Y(x))\} = \alpha + \beta x \tag{1}$$

The results from the above equation are reported as the log odds of an event occurring over not occurring. This can be further expressed as shown below:

$$Y(x) = \exp(\alpha + \beta x)/\{1 + \exp(\alpha + \beta x)\} \tag{2}$$

where Y is the dependent variable, X is the explanatory factor, $\alpha$ is the intercept and $\beta$ is the regression coefficient for X. The symbol e is the error term. When the explanatory factors are more than 2,

$$\beta x \text{ becomes } \beta_{1 \times 1} + \ldots \ldots + \beta_{m \times m}. \tag{3}$$

Therefore, a full ordinal regression for this study is expressed as shown below:

$$Y_1 = f(X_1, X_2, X_3, X_4, X_5, X_6, X_7) + e \tag{4}$$

Similar to the multiple logistic regression, the odds ratio was generated after running the ".ologit" command in Stata. Table 3 below shows the dependent and independent variables included in the ordered logistic regression.

**Table 3.** Definition of the variables in the ordered logistic regression.

| | Variables | Units of Measure |
|---|---|---|
| Y | Level of integrating soil health options | 1 = Low, 2 = Moderate, 3 = High |
| $X_1$ | Soil health options are beneficial to soils | 0 = No, 1 = Yes |
| $X_2$ | Lack of farm tools constrains soil health options | 0 = No, 1 = Yes |
| $X_3$ | Climate variability constrains soil health options | 0 = No, 1 = Yes |
| $X_4$ | Local leaders approve soil health options | 0 = No, 1 = Yes |
| $X_5$ | Level of ability to apply soil health options | 1 = Low, 2 = Moderate, 3 = High |
| $X_6$ | Sex of farmer | 0 = Female, 1 = Male |
| $X_7$ | Wealth status of farmer | 1 = Very poor, 2 = Poor, 3 = Better-off |

## 3. Results

### 3.1. Design of Learning Plots Managed by Smallholder Farmers

3.1.1. Size and Number of the Learning Plots

Results revealed variations in the size of plots managed by the farmers. A majority of the farmers in the FRTs and FRNs tested different soil health options on sub-plots (0.01 ha or 10 ridges by 10 m length). Therefore, they allocated a small portion of their farm to the experimentation (mean = 0.03 ha and 0.04 ha, respectively). On the contrary, the farmers in the Lead Farmer- and FFS-based interventions allocated a significant ($p < 0.001$) proportion of their land (mean = 0.2 ha and 0.4 ha, respectively). There were also differences in the number of sub-plots allocated to different soil health options within a learning site. For instance, while most of the farmers in the FRNs (84%) worked with three or more sub-plots, most of the lead farmers (81%), FRTs (64%) and all FFSs had two sub-plots.

3.1.2. Types of Soil Health Options Tested

Results from the observation of the plots revealed a wide range of innovations related to inorganic and organic resources, germplasm, plant population as well as physical land-conservation structures (Figure 1). However, the most popular options tested by farmers in all the interventions included planting patterns for maize and legume crops (e.g., single seed per station in maize and double rows per ridge in soya beans). Testing different rates of mineral fertilizer (full or 92 kg N/ha, half or quarter rates) was also common, especially among farmers participating in the Lead Farmer-, FRN- and FFS-based interventions. In all

the interventions, inorganic soil inputs were applied using bottle tops or cups numbered 8, 5 and 2 (equal to 8 g, 5 g and 2 g, respectively). Testing improved maize and legume varieties recommended by formal research organisations (promoted for their high yield, early maturity, and tolerance to pests, diseases and droughts) was another common option in all the interventions. Other popular soil health options tested in all the interventions included managing crop residues, applying animal and compost manure, as well as rotating maize with a sole legume crop (i.e., soya, groundnuts or pigeon peas).

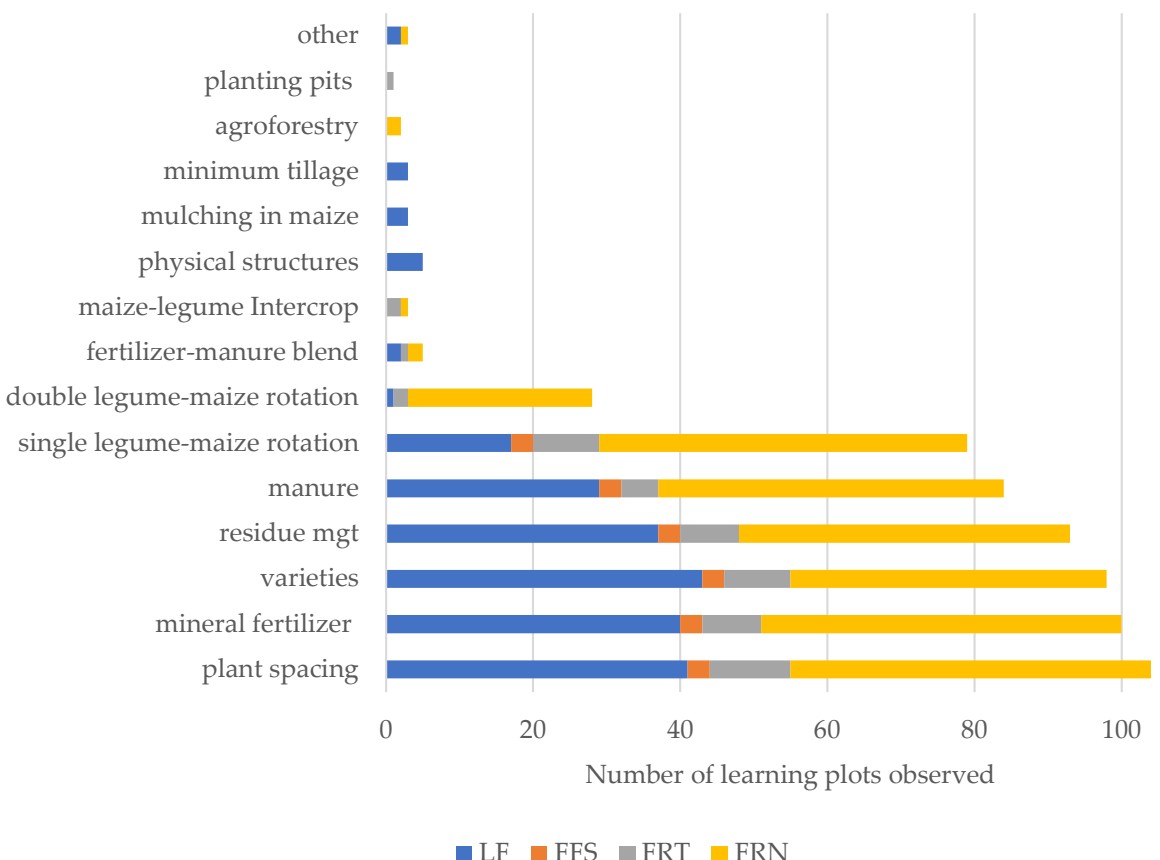

**Figure 1.** Types of soil health options tested by farmers in learning plots. n: LF = 43, FFS = 4, FRT = 11, FRN = 51.

There were also other soil health options tested in specific interventions. Rotation of maize with a double-up legume (i.e., growing two legumes in one field: pigeon peas plus groundnuts or soya, and groundnuts plus soya), for example, was tested only by farmers in the FRNs and FRTs. Only lead farmers tested physical structures (e.g., box ridges) as well as minimum tillage and mulching (i.e., conservation agriculture). Similarly, a few FRT members tested planting pits for conserving soil moisture, but this option was not tested in the Lead Farmer, FFS- and FRN-based interventions.

### 3.1.3. The Layout of the Learning Plots

The layout of the learning plots in all the interventions followed a side-by-side comparison where sub-plots treated with distinct soil health innovations were compared (Table 4). There was a sub-plot treated with an option recommended by researchers and another with the current practice followed by the farmers in the community. Earlier in this section, it was indicated that most of the farmers in the FRNs had more sub-plots than farmers in other interventions. These additional plots had treatments that combined farmer knowledge and the recommendations from the researchers (adaptation). Other FRN farmers also

added a sub-plot in which they tested their innovations. A few farmers involved in the Lead Farmer-, FRT- and FFS-based interventions worked on a single plot. In such cases, they compared a specific soil health option with practices applied in nearby fields within the village.

**Table 4.** Summary of soil health options compared in the learning plots.

| | |
|---|---|
| 1.　A typical layout of plots in the Lead Farmer- and FFS-based interventions (2–3 sub-plots) | |
| Plot 1: New soil health option from research (e.g., improved varieties, conservation agriculture—minimum tillage and mulching). | Plot 2: Current and popular soil health option (e.g., local variety, no conservation agriculture-field with ridges and no mulching). |
| 2.　A typical layout of plots in the FRT-based intervention (2–4 sub-plots) | |
| Plot 1: New soil health option from research (e.g., maize rotated with double-up legume). | Plot 2: Current and popular soil health option (e.g., continuous maize cropping with full-rate fertilizer application). |
| Plot 3: Soil health option adapted to the local context (e.g., composite/local maize variety and compost manure application). | Plot 4: Soil health option from research, which is familiar to farmers (e.g., Single legume rotated with maize). |
| 3.　A typical layout of plots in the FRN-based intervention (3–6 sub-plots) | |
| Plot 1: New soil health option from research (e.g., improved varieties, maize rotated with double-up legumes). | Plot 2: Current and popular soil health option (e.g., full-rate fertilizer application and continuous maize cropping). |
| Plot 3: Soil health option adapted to the local context (e.g., maize plot applied with quarter-rate fertilizer mixed with or following compost manure application). | Plot 4: Untested soil health option innovated by farmers (e.g., different rates of urine as an organic source of soil nutrients). |
| Plot 5: Soil health option from research that is familiar to farmers (e.g., single legume rotated with maize). | |

*3.2. Farmer Interactions in the Learning Plots*

The farmers interacted in the plots using various methods. However, meetings held on-site were the common mode of interaction. The farmers also met during the execution of various field activities (e.g., planting, weeding and fertilizer application). During the meetings, farmers were observing, recording and then discussing the positives, problems and lessons identified in the learning plots. They also held village-level field days as well as village workshops (unique to the FRN approach). During these activities, which were open to all the community members, the farmers shared their experiences and received feedback from other farmers (i.e., related to the performance of soil health options tested in the plots). These village forums were also spaces where farmers interacted with other interested stakeholders, including extension, research, university and NGOs operating in the areas.

In all the interventions, the frequency of interaction in the learning plots varied depending on the crop stage. The number of interactions ranged from one to three times a week during the stages of planting, germination and fertilizer application in maize. When the crop was well established, during the late vegetative stage, the farmers would usually meet once a fortnight. During the maturity stage of the crop, the frequency of interaction reduced to as low as once a month. Figure 2 shows farmers interacting in the learning plots.

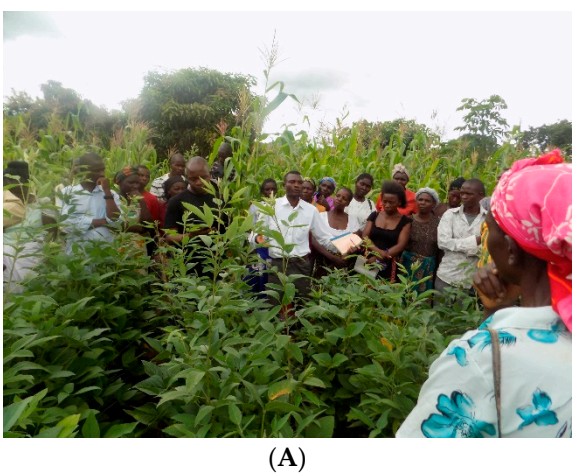
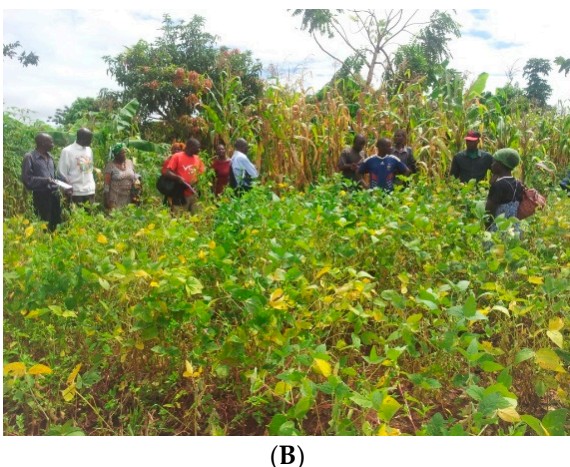

**(A)**　　　　　　　　　　　　　　　　　　　　　　　　**(B)**

**Figure 2.** Farmers, extension workers and agronomists interact in the learning plots. (**A**) FRT member sharing her observations with farmers in Zombwe EPA; (**B**) FRN joint field learning in Zombwe EPA.

*3.3. Farmer Perceptions of the Performance of Options Tested in the Learning Plots*

In all the interventions, farmers evaluated the different soil health options by observing the growth of the maize crop (i.e., maize response to soil health innovations). The results in Figure 3 show that whilst the majority of the farmers had favourable opinions towards the maize response to soil health options (very good and good), there were other farmers who felt that the options just performed reasonably or even poorly and very poorly. For example, out of the 45 farmers who evaluated groundnut/maize rotation for improving soil fertility and subsequent maize productivity, four indicated that it was a poor option. These unsatisfied farmers indicated that the maize from their plots had small cobs and ears.

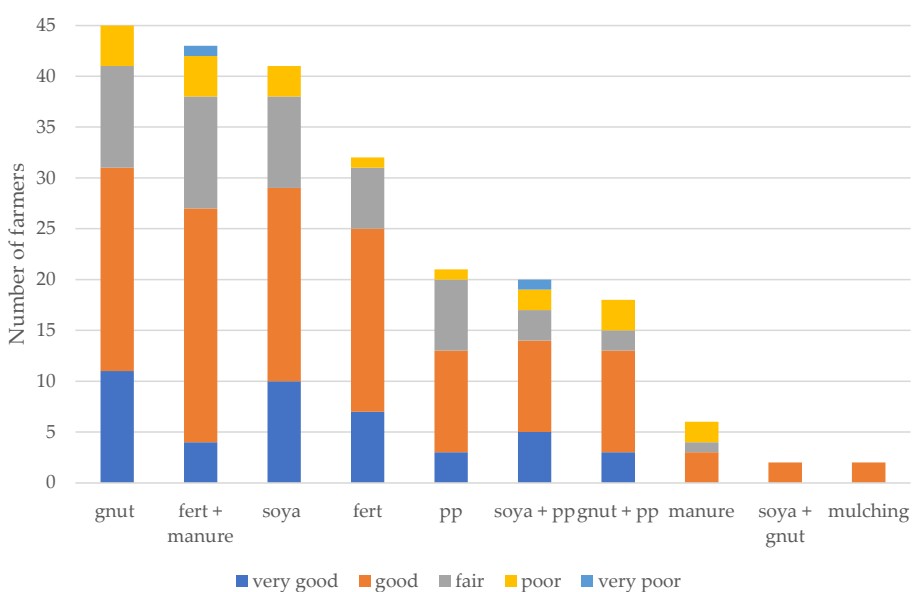

**Figure 3.** Farmer evaluation of the maize yield in selected soil health options. gnut = groundnuts-maize rotation, fert + manure = mineral fertilizer plus manure in continuous maize cropping, soya = soya-maize rotation, fert = mineral fertilizer in continuous maize cropping, pp = pigeon peas-maize rotation, soya + pp = soya plus pigeon peas-maize rotation, gnut + pp = groundnuts plus pigeon peas-maize rotation, manure = manure in continuous maize cropping, soya + gnut = soya plus groundnuts-maize rotation, mulching = soil cover using maize residues; all plots were incorporated with crop residues during land preparation and had improved varieties; n = 109.

Figure 4 shows a case where variation in maize response was observed in different plots implemented by a male and female farmer participating in the FRN. The maize looked different even though the two farmers planted the maize in the same village and treated the different plots with the same soil health option.

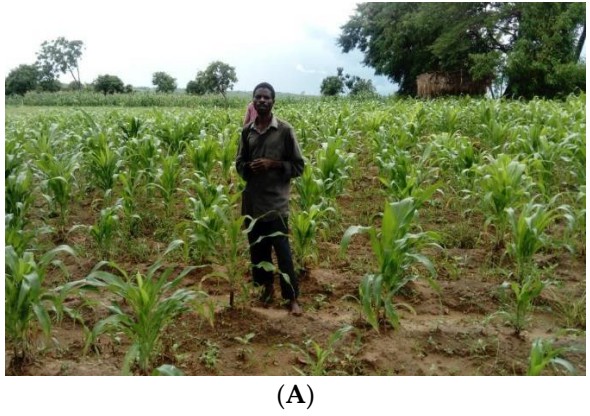 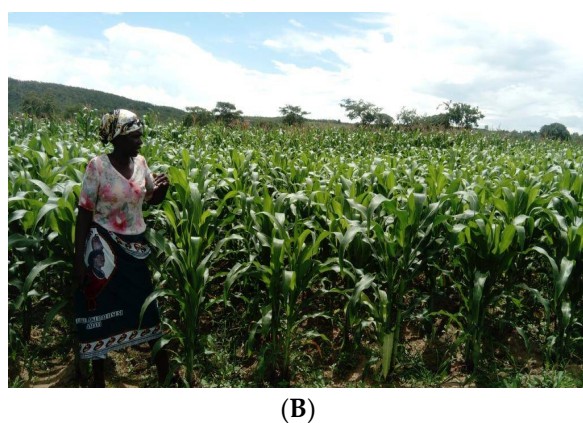

(**A**) (**B**)

**Figure 4.** Maize response to groundnut rotation as a soil health option: (**A**) Male FRN farmer in poorly performing plot (Kaunda Village); (**B**) Female FRN farmer in well-performing plot (Kaunda Village). In both plots, maize was rotated with groundnuts, and both plots are in the same village.

The multiple logistic regression was run to establish the sources of variations in the performance of the soil health options. The likelihood ratio chi-square (17.01) in the regression output was significant ($p < 0.01$), indicating that the variables in the model fitted the regression line. Further, the significant linear predicted value (_hat) means that the model had meaningful predictors ($p < 0.05$). The insignificant linear predicted value squared (_hatsq) also confirmed the absence of specification errors in the regression. Further investigation on the association between the independent variables revealed insignificant chi-square $p$-values for all possible combinations, indicating the absence of multicollinearity.

The results in Table 5 show that almost all the independent variables were significant. On the one hand, the odds ratio of having a good maize response to soil health options reduced significantly when the learning plots were subjected to runoff, waterlogging conditions and moisture stress ($p < 0.05$). On the other hand, the chances of having a ratio in favour of good maize response over that of a crop failure increased when seeding was on time ($p < 0.01$). The odds of getting a good maize crop were reduced by 73% when there was moisture stress, by around 87% in waterlogged plots and by nearly 90% where farmers experienced runoff, whereas planting on time raised the chances by five and a half times, holding all other variables constant.

**Table 5.** Factors associated with maize response to soil health options.

| Maize Performance | Odds Ratio | Std. Err. | $p < 0.05$ |
|---|---|---|---|
| Loam soils | 0.792189 | 0.486245 | 0.704 |
| Run-off | 0.098251 | 0.106316 | 0.032 |
| Waterlogged | 0.125364 | 0.117053 | 0.026 |
| Moisture stress | 0.268807 | 0.175747 | 0.044 |
| Timely planting | 5.624732 | 3.542196 | 0.006 |
| Constant | 4.109399 | 2.561062 | 0.023 |

n = 103, Likelihood ratio chi2 = 17.01 ($p$-value = 0.0045), Pseudo R2 = 0.1782, coefficient for linear predicted value (_hat) = 1.74 ($p$-value = 0.021), coefficient for linear predicted value squared (_hatsq) = −0.26 ($p$-value = 0.274): chi-square $p$-values were not significant for all possible associations between the independent variables.

### 3.4. Scaling-Up of Options Tested in the Learning Plots

When farmers were asked to indicate the soil health options they had learned and applied in the main fields, results in Figure 5 show that farmers had acquired knowledge on a wide range of options. However, the proportions of farmers who indicated that they applied the options on their farms were less than the proportions who expressed having acquired knowledge through the interventions. Notably, significant gaps existed in options such as burying residues (47%), mineral fertilizer application and maize-legume intercropping (42%) as well as pit planting, land resources structures and minimum tillage (36%, 34% and 34%, respectively). Surprisingly, despite being new, around one-third of those few farmers who indicated having acquired knowledge about double-up legume had also applied the option in their fields. Observations in the fields revealed that only farmers in the FRNs and FRTs were applying the double-up legume option.

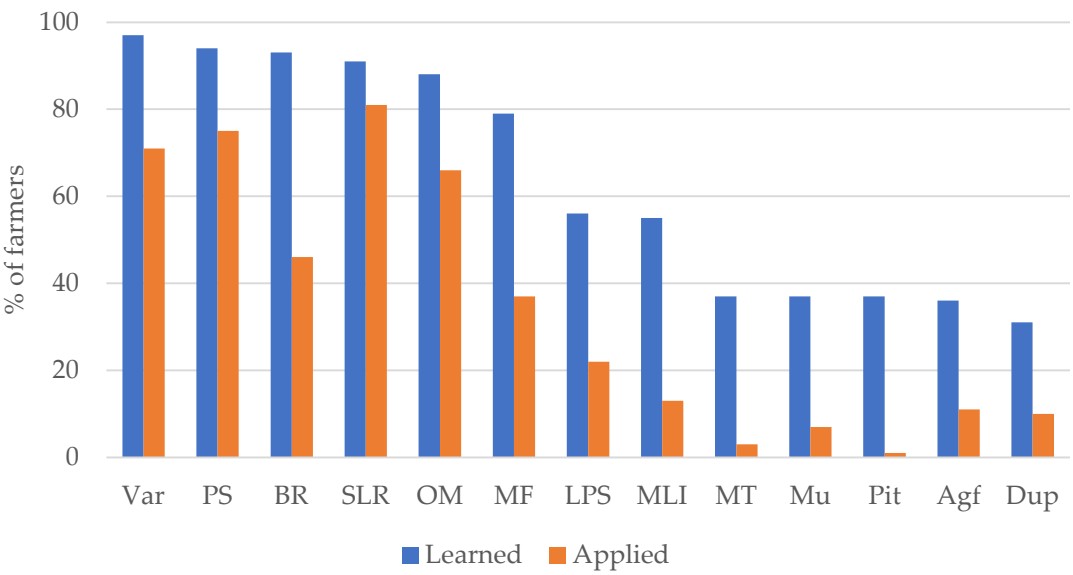

**Figure 5.** Percentage of farmers who learned and applied soil health options. Var = varieties, PS = plant spacing, BR = burry residues, SLR = sole legume rotation, OM = organic manure, MF = mineral fertilizer, LPS = land resources physical structures, MLI = maize legume intercropping, MT = minimum tillage, Mu = mulching, Pit = pit planting, Agf = agroforestry, Dup = double-up legume; Percentages generated from multiple response analysis; n = 197.

### 3.5. Factors Associated with Farmers' Decision to Scale-Up the Tested Options

On the number of options, farmers applied to improve the soil health of their fields, the results revealed that the majority of the farmers combined several options. Around 45% incorporated more than four options, for example, burying residue, planting improved varieties, mixing manure and fertilizer, and rotating a legume crop with maize (i.e., high level of integration). Another 42% of the farmers incorporated three to four options such as appropriate plant spacing plus maize-legume intercropping and box ridges for conserving moisture and protecting the soils from water runoff (medium level of integration). The remaining 13% applied at most two options in their fields and mainly included the improved varieties and full-rate mineral fertilizer (low level of integration).

Further analysis was conducted to explain the variations in the number of soil health options farmers had decided to integrate into their fields. The analysis involved regressing the level of soil health options integration (high, medium and low) with socioeconomic characteristics of farmers (Table 6).

**Table 6.** Factors associated with the level of farmers' integration of soil health options.

| Level of Soil Health Options Integration | Odds Ratio | Std. Err. | $p < 0.05$ |
|---|---|---|---|
| Options are beneficial to soils | 1.868201 | 0.590174 | 0.048 |
| Lack of farm tools constrains options | 0.705633 | 0.414876 | 0.553 |
| Climate variability constrains options | 1.271987 | 0.385598 | 0.427 |
| Local leaders approve options | 10.96507 | 6.878881 | 0.000 |
| Level of ability in applying options | | | |
|   Moderate | 0.94048 | 0.485724 | 0.905 |
|   High | 2.038556 | 1.094735 | 0.185 |
| Sex of farmer in the intervention (male) | 0.354341 | 0.11508 | 0.001 |
| Farmer wealth status in the village | | | |
|   Poor | 2.27837 | 0.748871 | 0.012 |
|   Very poor | 3.04280 | 1.472555 | 0.021 |
| Cut/1 | 0.535664 | 0.746611 | |
| Cut/2 | 3.217712 | 0.795405 | |

Number of observations = 195; Likelihood ratio chi-square = 50.92 (*p*-value = 0.0000); Likelihood-ratio test of proportionality of odds across response categories = 6.2 (chi2 *p*-value = 0.5168); Brant Test of Parallel Regression Assumption, for all variables = 7.1 (chi2 *p*-value = 0.418); Pseudo R2 = 0.1332.

In the regression output, the likelihood ratio chi-square was significant ($p < 0.001$), indicating that there was a significant difference between the model without the predictors and the one with the socioeconomic variables added. Further, the tests of the proportionality of odds across the response categories and the parallel regression assumption were both not significant. Therefore, the full regression was regarded as well-fitted. The proportional odds assumption was also not violated.

The results show that several socioeconomic variables were significantly associated with the level of integration. The socioeconomic variables related to the level of farmers' integration of soil health options included perceptions held by farmers regarding the benefits of the options ($p < 0.05$), approval of the options by local leaders ($p < 0.001$) as well as their gender ($p < 0.01$) and wealth status ($p < 0.05$).

The odds ratio of getting a farmer with a high level of integrating various soil health options almost doubled when farmers perceived that the different options were beneficial to soil health improvement. This was a popular opinion expressed by the farmers during the field visits. The farmers explained that they had combined several options, including organic resources, because it was helping improve the soil structure, moisture retention as well as reduced Striga (witchweed).

The chances of a farmer integrating several soil health options also increased by almost 11 times where influential individuals such as local leaders approved the application of the agro-ecological options in their areas. For instance, in Mkanakhoti EPA, it was learned that the village heads and traditional authorities encouraged their subjects to apply different soil health options to combat food insecurity and poverty. On the contrary, there were cases where the farmers felt that their village members were discouraging them from practicing the options and participating in soil health-related groups. For example, one married female farmer complained that some village members told her to stop participating in the FFS activities. She recounted the words that were spoken by the village members as

*"We have heard that, in that group, women share tactics on sleeping around with other men and cheating on their husbands. Do not get shocked, if he (husband) gets another woman. It is because of your unfaithful behaviour. Moreover, the solutions [soil health improving options] you learn from that group are a waste of time to your household. They never work. The only person benefiting is the village head, whose name is getting famous outside this village and to the Government".* (Female FFS member, face-to-face individual interview 116, April 2017)

Regarding gender, the likelihood of farmers integrating several soil health options was reduced by 35% when they were male. Likewise, the chances of the poor and very poor farm households applying a broad range of options were higher than for those who were better off in the community. The odds ratio was twice (for the poor) and three times (for the very poor) more than their well-to-do counterparts.

Observations on farmers' fields revealed that farmers had made some changes to the option. One or more features of the options applied by the farmers were different from those in the learning plots. Results from the survey confirmed that the majority (63%) of the farmers did not apply the options as demonstrated in the test plots. When the farmers were asked to explain why they had decided to make changes to the options, there were various reasons indicated by the respondents (Table 7).

**Table 7.** Why farmers made changes to soil health options.

| Recommendation From The Learning Plot | Changes Made in the Main Field | Reasons |
|---|---|---|
| Hybrid varieties for high residues and grain yield. | Plant a different variety, e.g., open pollinated or local variety. | - No access to hybrid seed due to lack of money and markets,<br>- Hybrid seed fails to perform in poor soils, e.g., waterlogged fields,<br>- Hybrid seed requires too much fertilizer, has no taste, low maize flour and susceptible to storage pests. |
| One seed per station for appropriate plant population. | Increase the number of seeds per station, e.g., 2–3 for maize or 2–5 for legumes. | - Security from loss of seed and seedlings due to low-quality seed, pests and dry spells,<br>- Requires less seed and fertilizer. |
| Maize spaced at 25 cm for appropriate plant population. | Increase between plants space, e.g., 30 to 75 cm for maize. | - Demands less labour during planting, weeding and fertilizer application, especially when alone, with children, old aged or when late with planting,<br>- Easy to intercrop maize with legumes, e.g., beans. |
| Intercrop maize with pigeon peas. | Intercrop maize with soya. | - The seed for soya is more available than other legume crops,<br>- Soya provides soil nutrients, feed, food and earns more income than pigeon peas. |
| Minimum soil disturbance plus soil cover. | Minimum tillage but no mulching. | - Loss of maize stalks due to livestock and fire. |
| Mulch maize with maize residues. | Mulch maize with legume residues, e.g., soya and groundnuts | - Legume residues add more nutrients to soil than maize stalks. |

## 4. Discussion

### 4.1. Do Farmers Have the Capacity to Engage in Rigorous Experimentation?

Previous scholars reported on smallholder farmers testing different innovations in Sub-Saharan Africa, for example, Hockett and Richardson as well as Sumberg and associates [34,35]. This study took a step further by looking at the design of the experiments implemented by farmers. The findings reveal that, despite the differences in the designs of the learning plots (e.g., area and number of sub-plots), the farmers followed some of the vital research principles. These included having comparison groups, treatments and replication. The learning plots had some form of a control treatment (mainly a plot with common practices in the community). Farmers in the FRNs demonstrated the ability to manage experiments with multiple treatments (options) within a learning plot (e.g., formal science compared to local adaptation and indigenous knowledge). These treatments were given similar management and compared similar fields. The replication principle was achieved in the FRNs and FRTs, where a considerable number of farmers (moderate N of 10-20) were testing similar soil health options in different fields within a village.

The smallholder farmers investigated in this study may have been able to design and manage the learning plots because of the training they were receiving from the agronomists and extension officers. In the Lead Farmer- and FFS-based interventions, the designs and procedures for managing the learning plots were passed on to farmers by extension workers or researchers (through training), whilst in the FRNs and FRTs, the process of designing a plot started with agronomists and extension workers asking farmers to share experiences on how they tested innovations in their farms. Once farmers' knowledge and practices were established, the agronomists shared with the farmers the basic principles of experimentation. The process served to fill the gaps in farmers' knowledge and skills in research. In the end, both the farmers and the external agricultural players (i.e., researchers and extension agents) engaged in a dialogue to agree on the management and design of the learning plot while giving weight to knowledge contributed by all actors.

The existing social capital (bridging and bonding) may have equally contributed to the rigour, especially where different socioeconomic categories of farmers worked together in designing and managing the learning plots (making use of bridging social capital). Within the FRT and FRN groups, for example, some farmers had been involved in previous research initiatives and, therefore, were acquainted with basic principles of experimentation. These experienced farmers helped other farmers in the groups in setting up and managing the experimental plots. In addition, the farmers monitored each member's plot to ensure adherence to the agreed research design. They also guided each other in data collection and recording and jointly evaluated the options at the end of the farming season. The importance of social capital was also observed in the Lead Farmer- and FFS-based interventions, where farmers collectively set up and managed the learning plots (making use of bonding social capital).

### 4.2. Does Experiential and Social Learning Influence Farmers' Perceptions of Soil Health Options?

Implementation of the learning plots allowed the farmers to have a hands-on experience with different soil health management options for four months (full rainy season). It is likely that this season-long experiential learning opportunity enabled farmers to observe, reflect and make judgments regarding the suitability and application of the soil health options to their contexts. The learning plots also gave the farmers a platform for accessing agricultural extension services (e.g., new information and improved seeds) and for engaging in social learning (knowledge sharing between farmers, extension workers and agronomists).

The interaction of diverse actors helped to increase farmers' awareness of the different soil health options as well as exposed them to diverse experiences and opinions about the performance of the options. For instance, in the FRTs and FRNs, one farmer would share experiences on how well a particular soil health option performed in his or her plot. In contrast, another farmer would share the shortfalls of the same option in the same village. The agricultural researchers and extension agents would then contribute to the

discussion by explaining the possible causes of the variations in the performance of the option. Gradually, such discussions allowed farmers to evaluate and form perceptions about the soil health options based on information from multiple realities. Scholars also reported a similar learning process in co-learning cycles in Mali [36].

*4.3. How Farmers Form Perceptions of Soil Health Options: Does the Nexus of Biophysical and Socioeconomic Factors Matter?*

Scholars are recognising that innovations do not suit all farmers' contexts and, therefore, caution should be taken when promoting innovations among smallholder farmers on a wide scale. These scholars call for the application of farmer-centred approaches such as the FRNs, which allow farmers to test options in different contexts, a concept they call "Option by Context (OxC)" [29]. In this study, the learning plots implemented by farmers participating in interventions that applied different farmer-centred approaches (Lead Farmer, FFS, FRT and FRN) also demonstrate how the performance of soil health options varied among different farmers and locations. The literature also provides evidence of instability in the performance of soil fertility innovations in different contexts. In Malawi, despite showing promising results in other sites, the soil fertility improving options based on maize-legume intercropping struggled to perform in the steep slopes of the Songani watershed area. It was not possible to increase yield in the area due to eroded soils and the prohibitive cost of labour [37]. In addition, in Malawi, evidence was reported that showed variability in maize response to integrated soil fertility management options due to biophysical soil properties, weather and management [38]. These findings confirm the cautions stressed in the literature regarding the risks of concluding soil health experiments based on averages [39] and making blanket recommendations for farmers in diverse contexts [40].

As in previous studies, the soil health options tested by farmers in this study also failed to suit the diverse and unpredictable conditions experienced by farmers. The maize did not respond well where biophysical factors such as dry spells and too much rainfall were prevalent (causing run-off or waterlogging conditions). Such stressful conditions tend to contribute to crops being susceptible to problems such as parasitic weeds (Striga) and pests. Socioeconomic factors also played a significant role in determining the maize response to the options. For example, some of the learning plots were planted late probably because the farmers could not access the seed on time (especially where researchers and extension workers were late to deliver seed).

*4.4. What Influences Farmers' Decision to Scale-Out Soil Health Options?*

This study shows evidence of the increase in farmers' awareness and knowledge of soil health options as a result of participating in interventions that applied farmer-centred approaches. Despite the observation that the proportion of farmers who applied different options in their main fields was less than those who had knowledge of these options, the findings also revealed that the majority (87%) were integrating more than three soil health options in their main fields.

Mhango identified constraints related to biophysical, socioeconomic and cultural issues as limiting the uptake of legume-based soil improvement innovations in Northern Malawi [41]. In this study, the farmers' perceptions of the various socioeconomic factors prevailing in their context were crucial when taking soil health options to scale. The farmers scaled-out the options when they viewed them as beneficial to their farms and households as well as acceptable to their leaders (referents) and peers.

Gender is also a critical factor. The findings that female farmers were likely to scale-out more options than the males may be attributed to their differentiated gender roles in the households. In developing countries, the female household members tend to assume both reproductive (e.g., childbearing and care) and productive (e.g., domestic and agricultural activities) roles, especially when the households experience shocks [42]. In this study, female farmers opted for diverse options that would reduce expenditure on farm inputs but

ensure optimal yield, food and nutrition security. By contrast, the male farmers preferred intensive inputs, thus they could increase productivity and make the much-needed cash.

In practice, female farmers mixed different options such as low dosage of fertilizer plus manure and integrating different legumes such as groundnuts, soya and pigeon peas in their maize farming system. On the other hand, male farmers mainly invested in mineral fertilizer to maximise the maize yield per hectare. Other studies in Malawi have also found more female than male farmers preferring legume-based soil-improving options [43,44].

Previous studies have shown that the wealth status of farmers is also an important factor associated with the scaling of soil health options. For instance, Kamanga concluded that the low-resourced categories of farmers in Chisepo area of Malawi invested insufficiently in soil fertility management technologies [45]. Conversely, another study in Mozambique concluded that the intentions to apply conservation agriculture were high among the most impoverished farmers [46]. The results of this study show that the less-endowed farmers were likely to scale-out diverse soil health options to compensate for the deficit in the number of purchased soil inputs. The resource-poor farmers could not afford the mineral fertilizer. Consequently, they opted for mixing different options such as compost manure, crop residue incorporation, crop rotation, intercropping, agroforestry and a low dosage of mineral fertilizer. By contrast, the better-off farmers could afford to buy fertilizer or had enough livestock to supply the amount of manure required in their fields. Therefore, the better-off farmers did not need to look for different soil health options; one or two were enough.

Finally, it is clear from the results of the study that, while farmers were applying the different soil health options, most did not reproduce the knowledge acquired through the interventions in which they participated. Instead, the farmers partially applied the components of different options to improve the soil health on their farms. The farmers opted for adapting different soil health options to local conditions rather than replicating the acquired knowledge. Scholars have referred to such integration and adaptation of different soil health improvement options as an advanced level of managing farm soils (complete integrated soil fertility management) [47].

**Author Contributions:** Conceptualization, F.T., K.W. and J.M.; methodology, F.T., K.W. and J.M.; validation, D.K. and K.W.; formal analysis, F.T.; investigation, F.T., D.K., D.M. and W.M.; writing—original draft preparation, F.T.; writing—review and editing, K.W., J.M., D.K. and W.M.; visualization, F.T. and Daniso Mweu. All authors have read and agreed to the published version of the manuscript.

**Funding:** This research was funded by Collaborative Crop Research Programme of the McKnight Foundation, grant number 14-247.

**Institutional Review Board Statement:** NRI Project B0430 (000900432); Approval date: 1 December 2015; Ethics Committee: Chairman of the Research Ethics Committee/Faculty Research Degree Committee, University of Greenwich; "Having satisfied all the relevant ethical and regulatory requirements, I am pleased to inform you that the above referred research protocol has officially been approved. You are now permitted to proceed with its implementation."

**Informed Consent Statement:** Informed consent was obtained from all subjects involved in the study.

**Data Availability Statement:** Data are not publicly available, though the data may be made available on request from the corresponding author.

**Acknowledgments:** The authors wish to thank farmers and field officers from Kandeu, Mkanakhoti and Zombwe EPAs for their commitment during data collection, and the Bunda College's Maize-Legume Best Bets Technologies project team for the technical support.

**Conflicts of Interest:** The study was financially supported by the McKnight Foundation through the Collaborative Crop Research Programme. The programme promotes the use of FRNs, one of the approaches analysed in this study. This would have been a source of bias in that funder could influence the researchers to collect and analyse the data in favour of the Farmer Research Networks. However, this threat was addressed by including in the research team, representatives of organisations that were using the other approaches focused on the study, namely the Lead Farmer, FFS and FRT. In

addition, the McKnight Foundation had no role in the design of the study; in the collection, analyses or interpretation of data; in the writing of the manuscript, or in the decision to publish the results.

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
