# Peer review of "From Learning Plot to Main Field: Scaling-Out Soil Health Innovations in Malawi"

_sustainability, doi:10.3390/su14031532_

Round 1

Reviewer 1 Report

The article is interesting and actual. It will be of interest to the readers.

The article is fine and just some revisions are need.

For example:

1) the article needs a deep formating effort, because many section titles are not in accordance with the template of the journal. For axample, lines 259, 260, 270, and so on.

2) Are the colors used in Table 4 really needed?

3) The quality of Figure 2(a) is poor. Use at least a 150 dpi picture.

4) Figure 4, the "B" must be "b".

Reviewer 2 Report

Dear Authors,

I had the opportunity to read and review the manuscript entitled „From Learning Plot to Main Field: Scaling-Out Soil Health Innovations in Malawi” (ID Sustainability-1534428).

The manuscript aims to examine how smallholder farmers in rural Malawi evaluated soil health options as well as how they scaled out their learning from the learning plots to their main farms.

The manuscript is interesting and fits within the scope of Sustainability: sustainable soil management requires engaging smallholder farmers in systematic experimentation in order to evaluate soil health options. I suggest some improvements in my review below.

Title: Informative and consistent with the study's content.

Keywords: There are a couple of keywords in the title already, so they need to be reconsidered.

Abstract: This abstract is clear and reasonable, contains the objectives of the study, data collection, and the main research findings, so it meets the journal's requirements.

  1. Introduction: It appears this section adequately supports the empirical research, emphasizing the importance of encouraging behavior change among smallholder farmers towards more productive and sustainable farming practices by involving them in the evaluation of soil health options.
  2. Methods (according to the journal’s requirements: Materials and methods): Data collection and analysis methods are clearly described and complete, as are the types of data collected.

In line 242, please check equation (1), which is really equation (3): the minus sign is left, 1-Y (x).

In line 245, please check equation (2), which is really equation (3): no dot is required after 'exp'.

Table 3: There is an inconsistency in the coding of the levels for ordinal variables: for variables Y and X5 the lowest level is coded with 1, but for variable X7 (wealth status of farmers) the highest level is coded with 1. The same goes for coding farmers ’opinions (lines 173-174).

I miss the description of the logistic model, the results of which are shown in Table 5.

  1. Results: the structure and content of this section are clear, but some minor corrections are needed.

For four participatory approaches, Figure 1 shows the number of farmers for each type of soil health option, please check the title of the Figure and the sentence in line 292.

Figure 3: the maximum value of the value axis can be set to 45. What do you mean by "good number" of farmers (lines 341-342)?

Please explain the variables in the multiple logistic model that was run to determine the sources of variations in the performance of soil health options (Table5): provide the levels of the dependent variable (maize response to soil health options) and the levels or categories of predictor variables (independent variables). What are the reference categories in the case of independent variables?

Table 5: since p-values are indicated, the notation (* p <0.05, ** p <0.01) is unnecessary.

The interpretations of Odds Ratios are incorrect: it's not possible to represent an odds ratio as a simple percent increase or decrease of an event occurring! E.g. the correct interpretation of the Odds Ratio corresponding to Moisture stress (OR=0.27): the odds of getting a good maize crop are reduced by 73% when moisture stress was present (it is 73% less likely to get a good maize crop if moisture stress is present).

According to my opinion, the type of participatory approaches (LF, FFS, FRT, FRN) should be taken into account in constructing a logistic model for examining the effects of factors associated with levels of farmers' integration of soil health options.

Line 437: This section…

Table 6: since p-values are indicated, the notation (* p <0.05, ** p <0.01) is unnecessary.

Overall, the evaluation of the results is correct.

  1. Discussion: The fundings of the study were compared with the results of previous research. In my opinion, the evaluation of the results is correct.

5. Conclusion: Because there is some overlap between this section and the abstract, it is not absolutely necessary.

Reviewer 3 Report

Dear Authors

The paper is well written and clearly articulated. I have made some minor suggestions in the way of vocabulary, and layout. The main area where I suggest improvement is the connection of the learning and field plot to actual interview. I would like to see explanation of human ethics clearance and whether this was sought or gained. I have attached a marked up pdf which elaborates on the areas I have suggested. On line 109 make sure the number of interviews is clear and how interviewed. Does this only occur at the site visit. The number of plots visited is not necessary equal to the number of interviews undertaken. In terms of data presentation can figure 1 the data be standardised so able to compare unequal sample sizes? We have the actual numbers in a table earlier but hard to compare if some have a greater % sampled. Many of the remaining comments are english suggestions. In the discussion I have highlighted where the statements need to be grounded more in the data and supported by the actual study findings.
